# C-C motif chemokine receptor 2 and 7 synergistically control inflammatory monocyte recruitment but the infecting virus dictates monocyte function in the brain
Clayton W. Winkler [1] ✉, Alyssa B. Evans [1,4], Aaron B. Carmody[2], Justin B. Lack[3], Tyson A. Woods[1] & Karin E. Peterson[1]

Inflammatory monocytes (iMO) are recruited from the bone marrow to the brain during viral encephalitis. C-C motif chemokine receptor (CCR) 2 deficiency substantially reduces iMO recruitment for most, but not all encephalitic viruses. Here we show CCR7 acts synergistically with CCR2 to control this process. Following Herpes simplex virus type-1 (HSV-1), or La Crosse virus (LACV) infection, we find iMO proportions are reduced by approximately half in either *Ccr2* or *Ccr7* knockout mice compared to control mice. However, *Ccr2/Ccr7* double knockouts eliminate iMO recruitment following infection with either virus, indicating these receptors together control iMO recruitment. We also find that LACV induces a more robust iMO recruitment than HSV-1. However, unlike iMOs in HSV-1 infection, LACV-recruited iMOs do not influence neurological disease development. LACV-induced iMOs have higher expression of proinflammatory and proapoptotic but reduced mitotic, phagocytic and phagolysosomal transcripts compared to HSV-1-induced iMOs. Thus, virus-specific activation of iMOs affects their recruitment, activation, and function.

Leukocyte infiltration into the brain is associated with La Crosse virus (LACV)-induced encephalitis in humans[1]. LACV is the leading cause of pediatric arboviral encephalitis in North America and causes dozens of clinical cases annually[2], some of which can have severe outcomes[3]. Mice infected with LACV exhibit a similar age-dependent predisposition to encephalitic disease as humans, which lessens as the peripheral immune response develops while aging[4,5]. In weanling mice infected with a lethal dose of LACV, leukocytes were also found associated with disease[6,7]. Flow cytometry analysis identified inflammatory monocytes (iMOs) as the largest subset of brain-infiltrating leukocytes suggesting these cells may play a role in disease[7] and may represent a therapeutic target.

Hematopoietic progenitors in the bone marrow give rise to iMOs released into the blood in response to virus infection[8]. iMOs travel to sites of infection, infiltrate infected tissues and influence pathogenesis either directly[9] or by maturing into effector cells that contribute to the immune response[10]. Appropriate recruitment of iMOs is critical during viral encephalitis to effectively clear virus from the brain, while not damaging neurons. Viral pathogenesis studies have shown that iMOs can have ameliorating[11–14] or exacerbating effects on disease[15,16], depending on the pathogen. Despite their potential as a therapeutic target to treat disease, our understanding of the mechanisms involved in the recruitment of iMOs from the bone marrow and their activation state once they reach the brain, remains incomplete.

[1]Neuroimmunology Section, Laboratory of Neurological Infections and Immunity, Rocky Mountain Laboratories, Department of Intramural Research, National Institute of Allergy and Infectious Diseases, National Institutes of Health, Hamilton, MT 59840, USA. [2]Research Technologies Branch, Rocky Mountain Laboratories, Department of Intramural Research, National Institute of Allergy and Infectious Diseases, National Institutes of Health, Hamilton, MT, USA. [3]NIAID Collaborative Bioinformatics Resource, National Institutes of Allergy and Infectious Diseases, National Institutes of Health, Bethesda, MD, USA. [4]Present address: Department of Microbiology and Cell Biology, Montana State University, Bozeman, MT, USA. ✉e-mail: winklercw@niaid.nih.gov

One of the main initiators of monocyte recruitment is the signaling of the chemokine C-C motif ligand (CCL) 2 through the G protein-coupled chemokine receptor (GPCR) C-C motif receptor (CCR) 2[17,18]. For encephalitic viruses, CCL2 has been demonstrated to be produced in the periphery early in infection[19] and by neurons[20] or glial cells[7,21] later in infection. This signal must travel to the bone marrow where it is thought to activate CCR2-dependent desensitization of a CXCL12-CXCR4 stay signal in maturing monocytes[22] to facilitate their release into the blood. This role for CCR2 has been validated in multiple studies where $Ccr2^{-/-}$ mice have been infected with encephalitic viruses such as herpes simplex virus-1 (HSV-1)[11,12], West Nile virus[14] and Tahyna virus (TAHV)[7], and iMO recruitment to the blood and/or brain is impaired. Yet, $Ccr2^{-/-}$ mice infected with the encephalitic LACV or Jamestown Canyon virus, iMO recruitment was not impaired[7], suggesting an alternative pathway for iMO recruitment out of the bone marrow.

It is possible that multiple signaling pathways control virus-induced iMO recruitment, which could explain the variable dependence for CCR2 in this process. Cellular immune responses commonly integrate multiple signaling pathways to control both the intensity and timing of the response[23]. Additionally, different viruses may more strongly activate specific signaling molecules thus shifting the overall contribution of that molecule in the response[24,25]. Here we examined iMOs recruited to the brain during a CCR2-independent (LACV) versus a CCR2-dependent (HSV-1) infection for GPCR transcript expression. We found that $Ccr7$ transcripts and protein were more highly expressed on iMOs from LACV-infected mice. We then examined the role of both CCR2 and CCR7 in iMO recruitment in either single ($Ccr2^{-/-}$ or $Ccr7^{-/-}$), or double knockout ($Ccr2^{-/-} \times Ccr7^{-/-}$) mice during either LACV or HSV-1 infection and identified that both receptors contribute to iMO egress from the bone marrow and entry to the brain. Furthermore, we determined that LACV-recruited iMOs do not influence disease in contrast to HSV-1-recruited iMOs[11,12]. To address this difference, we transcriptomically analyzed LACV- and HSV-1-recruited iMOs and found differences in proinflammatory, phagocytic and phagolysomal pathways that could influence iMO function.

## Results

### CCR7 expression is increased on iMOs during LACV infection, but is not necessary for iMO recruitment to the brain

In previous studies, we found that LACV induced iMO recruitment from the bone marrow to the brain in the absence of CCR2[7], suggesting other receptors or signaling pathways may compensate for CCR2 in this process. To identify other receptors that might compensate for CCR2 in iMO recruitment, we compared transcriptional expression of an array of GPCRs, some of which are known to be involved in immune cell trafficking[26–29], in iMOs recruited to the brain during HSV-1 (CCR2-dependent iMO recruitment) and LACV (CCR2-independent iMO recruitment) infections. iMOs were isolated from the brains of three clinical HSV-1 and LACV infected mice by FACS (Fig. 1a–d), processed for RNA and then analyzed for GPCR transcriptional expression via RT² Profiler PCR Array (Fig. 1e). Multiple transcripts were significantly differentially expressed between iMOs isolated from LACV- versus HSV-1- infected mouse brains (Supplementary Data 1). The transcript with the largest fold change was $Ccr7$ (Fig. 1e, Supplementary Data 1), which has previously been linked to iMO migration to sites of parasitic infection[30]. CCR7 protein expression was also increased on iMOs from the brain of LACV infected mice relative to HSV-1 infected mice (Fig. 1f, g). However, in a previous analysis looking at different chemokine receptor knockouts, we had found that CCR7 deficiency alone did not inhibit iMO recruitment to the blood following a lethal dose of LACV ($10^3$ PFU)[7]. Similarly, when iMO recruitment to the brain was analyzed at the same clinical time point, no difference was observed in the percentage of iMOs in the brain between WT or $Ccr7^{-/-}$ mice during LACV infection (Fig. 1h). Thus, despite the increased expression of CCR7 by LACV-induced monocytes, CCR7 alone does not control iMO recruitment from the bone marrow to the brain during LACV infection.

### Generation of Ccr2⁻/⁻ RFP x Ccr7⁻/⁻ double knockout mice (DKO)

CCR2 and CCR7 may compensate for each other during LACV-induced iMO recruitment. To test this, we generated $Ccr2^{-/-}$ RFP $\times Ccr7^{-/-}$ double knockout (hereafter DKO) mice through selective breeding. Genotyping with specific primers to amplify wildtype and knockout gene segments identified mice that were either fully wildtype (WT), heterozygous for both Ccr2 and Ccr7 (hereafter HET) or DKO (Supplementary Fig. 1a, b). $Ccr2^{-/-}$ RFP mice have been reported to be monocytopenic under normal conditions[31]. To examine whether DKO mice were also monocytopenic, we compared iMOs in the blood of mock infected HET (Supplementary Fig. 1c–f) and $Ccr2^{-/-}$ RFP (Supplementary Fig. 1g–j) mice to DKO (Supplementary Fig. 1k–n) mice. DKO mice had a clear reduction in the iMO population compared to either group indicating that DKO mice were more severely monocytopenic even than $Ccr2^{-/-}$ RFP mice. DKO mice did retain other important blood cell types such as CD11c⁺ dendritic cells (Supplementary Fig. 1c, g vs k), and Ly6G⁺ granulocyte/neutrophils (Supplementary Fig. 1e, i vs m). Importantly, DKO mice had similar numbers of iMO progenitors in their bone marrow compared to HET mice indicating that the decrease of iMOs in the blood of DKO mice was not the result of impaired progenitor survival (Supplementary Fig. 2). The relatively few iMOs in the blood of DKO mice retained their RFP expression (Supplementary Fig. 1f vs n), confirming they were CCR2 deficient and could be tracked.

### Either CCR2 or CCR7 can mediate iMO recruitment to the blood during LACV infection but each receptor contributes additively to recruitment

To determine the individual contribution of CCR2 and CCR7 to iMO recruitment during LACV infection, $Ccr2^{-/-}$ RFP, $Ccr7^{-/-}$, HET and DKO mice were infected with a lethal $10^3$ PFU dose of LACV. iMOs were measured in the blood at different days post infection (dpi) (Fig. 2a–f). In HET mice, the proportion of iMOs in blood increased from a mock baseline during infection to a peak at 5dpi (Fig. 2a, e), followed by a return to near mock baseline at 7dpi (Fig. 2f). As previously reported, mock $Ccr2^{-/-}$ RFP mice were monocytopienic[17,32] relative to mock HET and $Ccr7^{-/-}$ mice. However, during infection, the proportion of iMOs increased in the blood of both $Ccr2^{-/-}$ RFP and $Ccr7^{-/-}$ mice at a similar rate to HET mice, but with only approximately half the magnitude (Fig. 2e, f). In contrast, there was no iMO recruitment in DKO mice, with no proportional increase in iMOs and levels lower than all other strains throughout infection (Fig. 2e, f). Thus, iMO recruitment was largely abolished suggesting that iMO recruitment from the bone marrow to the blood is strongly dependent on signaling through either CCR2 or CCR7 during lethal LACV infection, and that these receptors play a synergistic role to generate robust iMO recruitment.

### CCR2 and CCR7 contribute additively to iMO recruitment during HSV-1 infection

To determine whether CCR2 and CCR7 play a synergistic role in iMO recruitment for other encephalitic viruses, the proportion of iMOs in the blood (Fig. 2g) of $Ccr2^{-/-}$ RFP, $Ccr7^{-/-}$, HET and DKO mice was measured throughout HSV-1 infection after inoculation with $10^7$ FFU. There was an increase in iMOs in the blood of HET mice following HSV-1 infection, although considerably lower than that observed with LACV infection (Fig. 2e). The relative iMO percentages in each of the single knockouts were reduced compared to HET mice at all time points, with only a modest increase in iMO proportions in the blood at 3dpi that returned to near mock baseline by the 5dpi/clinical time point (Fig. 2g). Similar to LACV-infected mice, the proportions of iMOs in DKO infected mice was substantially lower than either single KO strain. Taken together, these data provide strong evidence that CCR2 and CCR7 play a synergistic role to mediate iMO recruitment during encephalitic virus infection, the latter of which has not been previously reported to be involved in iMO recruitment during viral infection.

## DKO mice have impaired iMO recruitment to the brain during LACV and HSV-1 infection

iMOs have been shown to influence disease during encephalitis or brain inflammation by migrating to the brain and exerting an effector function[9,11,12,14,32]. Thus, iMO recruitment to the brain of HET, single KO and DKO mice was examined at multiple time points during LACV infection (Fig. 3a-f). In mock-inoculated control mice, the proportion of iMOs were similar in the brains of HET, $Ccr2^{-/-}$ RFP and DKO mice but were

significantly elevated in $Ccr7^{-/-}$ mice by comparison (Fig. 3e). However, by 3dpi in LACV-infected mice, iMO infiltration had increased in $Ccr2^{-/-}$ RFP mice and were similar to $Ccr7^{-/-}$ mice. By 5 dpi, iMO proportions were increased in the brains of LACV-infected HET, $Ccr2^{-/-}$ RFP and $Ccr7^{-/-}$ mice, albeit with high variability. In contrast, iMO infiltration in brains of LACV-infected DKOs remained low. By the 7dpi timepoint (Fig. 3f), the majority of mice had developed clinical disease (solid symbols) with only a few non-clinical animals (open symbols) that had much lower iMO infiltration than

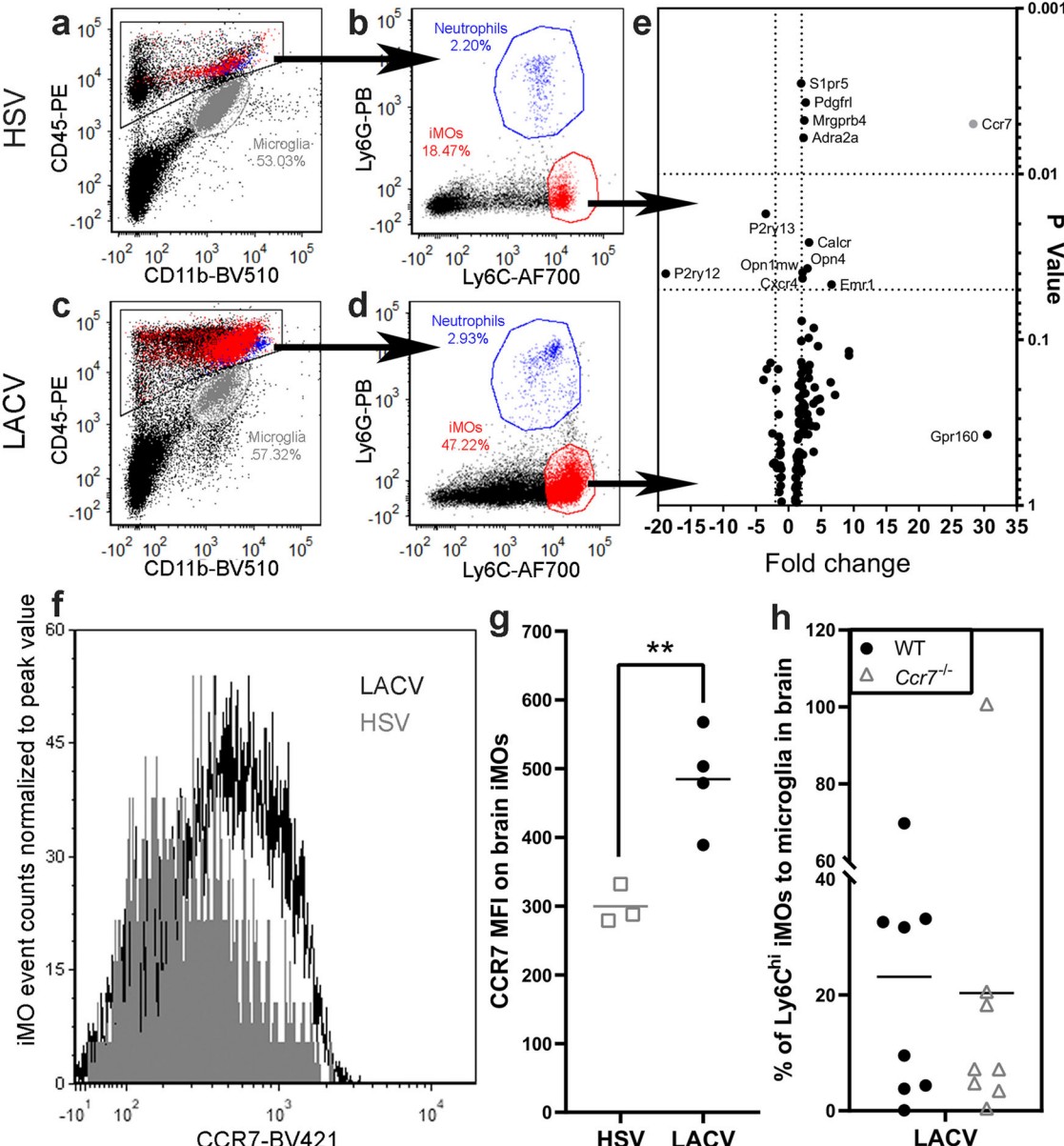

**Fig. 1 | CCR7 expression is elevated on iMOs from LACV infected mice, but CCR7 alone does not control iMO recruitment.** Representative FACs plots of iMO isolated (populations associated with the black arrows) from brains of (**a**, **b**) HSV-1 and (**c**, **d**) LACV IP infected mice. Gray = microglia, blue = granulocytes/neutrophils and red = iMOs. **e** Volcano plot of a RT² Profiler PCR Array of 370 GPCR transcripts from FACs isolated iMOs. Data are plotted as the expression fold change of LACV vs HSV-1 iMO transcripts (x-axis) relative to p-value as determined by a Student's *t* test between groups assuming equal variances (y-axis). Transcripts of highest significance and/or fold change are labeled and the complete data set is shown in Supplementary Table 1. n = 3 mice per group, all males. **f** Histogram of fluorescent intensity measurements of iMOs taken from the brain of LACV (black) and HSV-1 (gray)-infected, clinical mice. Data are normalized to peak values for ease of comparison due to the higher numbers of iMOs present in the brain of LACV

relative to HSV-1 infected mice. **g** The mean fluorescent intensity (MFI) of iMOs from the brains of clinical LACV (black circles) and HSV-1 (gray squares) mice are plotted. A two-tailed, unpaired t-test was used to examine differences in MFI between iMOs from LACV and HSV-1 clinical brains with the following statistics: t = 4.035, df = 5, p = 0.0100. ** indicates a p value ≤ 0.01. **h** The percent of Ly6c^hi iMOs relative to microglia in the brain of IP 10³ LACV-infected WT and $Ccr7^{-/-}$ mice at the 7dpi/clinical timepoint. This timepoint was selected to best correlate with the RT² Profiler PCR Array (**e**) and with peak of iMO infiltration into the brain[7]. Data point dispersion is likely accounted for by only ~50% of animals in each group reaching clinical disease. A two-tailed, unpaired t-test was performed to examine differences in iMO recruitment between WT and $Ccr7^{-/-}$ mice with the following statistics: t = 0.2226, df = 7, p = 0.8302. Results from individual animals are plotted with the black horizontal bar representing the mean.

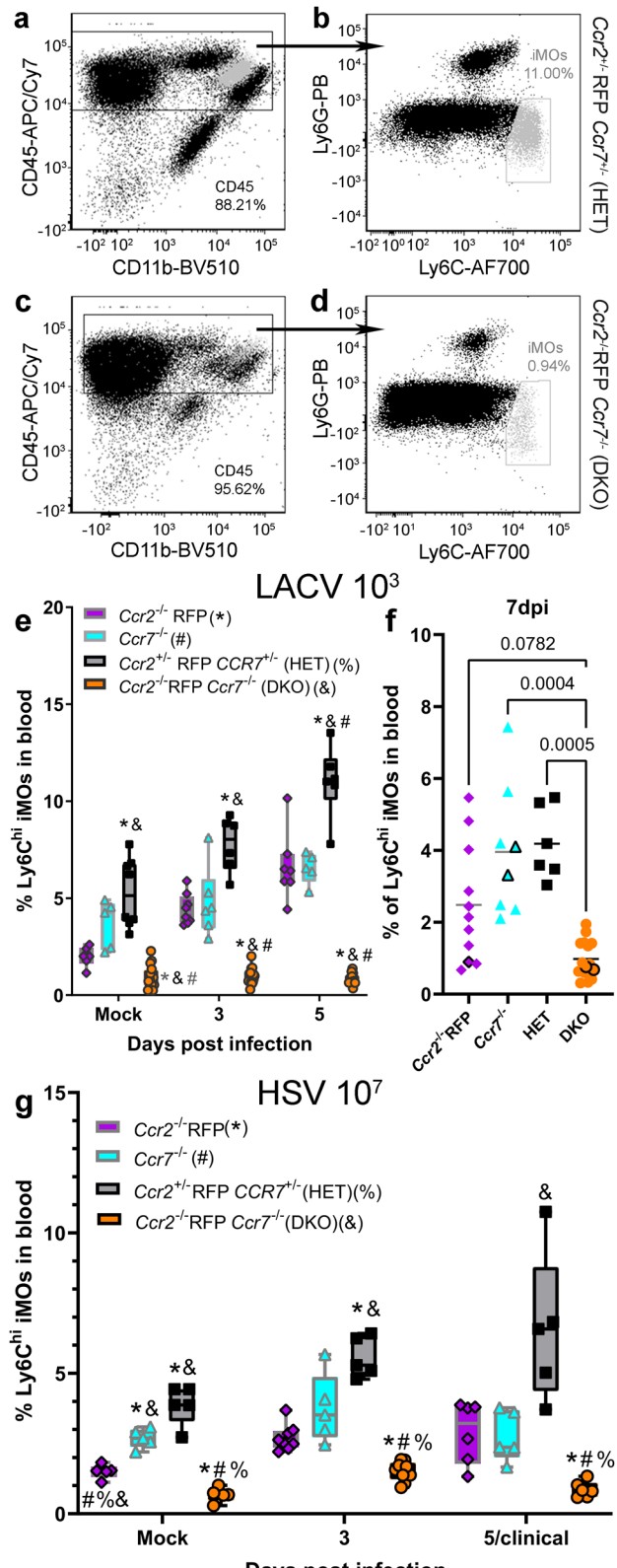

**Fig. 2 | DKO mice have impaired iMO recruitment to the blood.** Representative flow cytometric plots of Ly6C$^{hi}$, Ly6G$^-$ iMOs from the blood of mice with the highest (**a** and **b** HET) and lowest (**c** and **d**, DKO) iMO recruitment during 10$^3$ PFU LACV IP infection at 5dpi. The light gray labeled cells represent iMOs. **e** Time course analysis from mock to 5dpi of iMO recruitment to the blood of *Ccr2$^{-/-}$* RFP, *Ccr7$^{-/-}$*, HET and DKO mice infected IP with 10$^3$ PFU LACV. **f** iMO recruitment to the blood of *Ccr2$^{-/-}$* RFP, *Ccr7$^{-/-}$*, HET and DKO mice infected IP with 10$^3$ PFU LACV at the 7dpi/clinical time point. Data points with black boarders indicate nonclinical animals. All HET animals had clinical disease. **g** Time course analysis from mock to 5dpi of iMO recruitment to the blood of *Ccr2$^{-/-}$* RFP, *Ccr7$^{-/-}$*, HET and DKO mice infected IP with 10$^7$ FFU HSV-1. For **e–g**, data are presented in box-whisker plots with individual data points represented by each symbol within the plot for each mouse genotype. The plots are color-coded such that magenta represents data from *Ccr2$^{-/-}$* RFP, cyan from *Ccr7$^{-/-}$*, black/gray from HET and orange from DKO mice. The following numbers of mice were analyzed for iMO recruitment to blood (**e** and **f**) during lethal LACV infection: *Ccr2$^{-/-}$* RFP, $n = 6$ mock, $n = 8$ at 3dpi, $n = 7$ at 5dpi and $n = 11$ for 7dpi/clinical; *Ccr7$^{-/-}$*, $n = 5$ mock, $n = 6$ at 3dpi, $n = 5$ at 5dpi and $n = 8$ at 7dpi/clinical; HET, $n = 8$ mock, and $n = 6$ at 3, 5 and 7dpi/clinical; DKO, $n = 16$ mock and at 3dpi, $n = 14$ at 5dpi and $n = 15$ at 7dpi/clinical. For **g** HSV-1 infection, the following numbers of mice were analyzed for each genotype: *Ccr2$^{-/-}$* RFP mice, $n = 5$ mock, $n = 8$ at 3dpi and $n = 6$ at 5dpi/clinical. *Ccr7$^{-/-}$* and HET mice, $n = 5$ at each time point. DKO mice, $n = 5$ mock, $n = 9$ at 3dpi and $n = 6$ at 5dpi/clinical. For **e** LACV-infected and **g** HSV-1 infected mice, a two-way ANOVA mixed-effects analysis with a Geisser-Greenhouse correction was used to compare the proportion of iMOs in the blood of ($F_{(2.293,34.78)} = 14.35$, $p < 0.0001$) LACV and ($F_{(1.237,21.03)} = 10.61$, $p = 0.0023$) HSV-1 mice from mock to 5dpi with a Tukey's multiple-comparison test to compare four groups over multiple time points with an alpha of 0.05. Specific statistics can be found in Table 1. *, #, % or & symbols associated with a data point indicate the *p*-value for the statistically comparison between that data point and the time point-associated *Ccr2$^{-/-}$* RFP, *Ccr7$^{-/-}$*, HET or DKO data point respectively is below 0.05. Because not all mice developing clinical disease, a non-parametric Kruskal–Wallis test was used to compare the percentage of iMOs in the blood (**f**, Kruskal–Wallis statistic = 23.87, $p < 0.0001$) of *Ccr2$^{-/-}$* RFP, *Ccr7$^{-/-}$*, HET and DKO mice at the 7dpi/clinical time point with a Dunn's multiple comparisons test with an alpha of 0.05 to compare specific groups. *P*-values of comparisons to DKO mice are shown. All other statistics can be found in Table 1. Error bars represent SE for all groups. Dpi = days post infection.

lymph node (Supplementary Fig. 3a–d) at 3dpi when iMO recruitment to the blood is underway[7] and in the brain at the 3 and 5dpi time (Supplementary Fig. 3e–h) when iMO brain recruitment is increasing in all groups (Fig. 3e). Furthermore, at least one *Ccr2* or *Ccr7* ligand transcript was elevated in most infected mice relative to mock controls in both tissues demonstrating these recruitment signals were elevated despite receptor knockout. Thus, the lack of iMO recruitment to the brain in DKO animals does not appear to be due to a lack of recruiting signal.

The recruitment of iMOs to the brains of HET, *Ccr2$^{-/-}$* RFP, *Ccr7$^{-/-}$* and DKO mice was also examined during HSV-1 infection (Fig. 3g). Again, iMOs were elevated in the brains of mock *Ccr7$^{-/-}$* mice relative to all other groups (Fig. 3e), albeit without reaching statistical significance. During HSV-1 infection, the proportion of iMOs in the brain increased the most in HET mice at the 3 and 5dpi timepoints. This increase was not significantly different from single knockout mice but was significantly higher than DKO mice. The proportion of iMOs in the brain of infected HET mice was similar to *Ccr7$^{-/-}$* mice at 3 and 5 dpi although this may be influenced by the higher basal proportion of iMOs observed in *Ccr7$^{-/-}$* mice. iMO proportions in *Ccr2$^{-/-}$* RFP mice were intermediate of HET and DKO mice similar to observations in the blood (Fig. 2e, f, g). Collectively this data indicates that dual deletion of CCR2 and CCR7 strongly inhibits iMO recruitment to the brain during infection with either LACV or HSV-1.

## iMOs do not affect LACV pathogenesis

HSV-1 pathogenesis can be modulated by inhibiting iMO recruitment to the brain[11,12]. To determine if iMOs play a role in LACV-induced neuropathogenesis, HET and DKO weanling mice were infected with a lethal 10$^3$ PFU dose of LACV and followed for the development of neurological

clinical animals. At this time point, HET, *Ccr2$^{-/-}$* RFP and *Ccr7$^{-/-}$* mice had significantly higher proportions of iMOs in the brain relative to DKO mice (Fig. 3f). Thus, there was a lack of iMO recruitment to the blood and the brain in DKO mice, relative to single KO or HET mice during LACV infection.

The lack of iMO recruitment in DKO mice occurred despite similar levels of CCR2 and CCR7 ligand transcript expression in HET and DKO

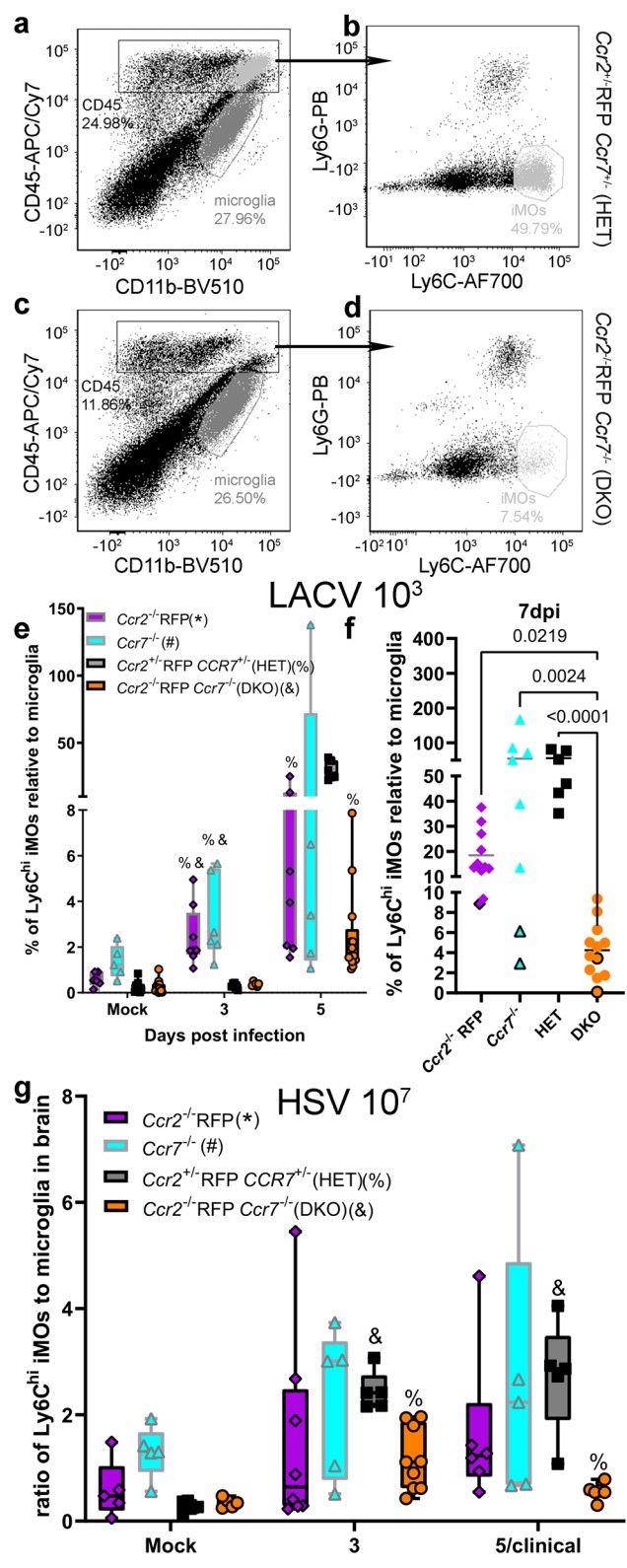

**Fig. 3 | DKO mice have impaired iMO recruitment to the brain.** Representative flow cytometric plots of Ly6C^hi, Ly6G^− iMOs from the brains of mice with the highest (**a, b** HET) and lowest (**c, d**, DKO) iMO recruitment during $10^3$ PFU LACV IP infection at 7dpi. The dark gray labeled cells represent microglia and the light gray represent iMOs. **e** Time course analysis from mock to 5dpi of iMO recruitment to the brain of *Ccr2*$^{−/−}$ RFP, *Ccr7*$^{−/−}$, HET and DKO mice infected IP with $10^3$ PFU LACV. **f** iMO recruitment to the blood of *Ccr2*$^{−/−}$ RFP, *Ccr7*$^{−/−}$, HET and DKO mice infected IP with $10^3$ PFU LACV at the 7dpi/clinical time point. Data points with black boarders indicate nonclinical animals. All HET animals had clinical disease. **g** Time course analysis from mock to 5dpi of iMO recruitment to the blood of *Ccr2*$^{−/−}$ RFP, *Ccr7*$^{−/−}$, HET and DKO mice infected IP with $10^7$ FFU HSV-1. For **e** LACV infection, the same *Ccr2*$^{−/−}$ RFP, *Ccr7*$^{−/−}$ and HET mice described in Fig. 2e, f were analyzed for iMO recruitment to the brain. Only a subset of the DKO mice used for blood analysis in Fig. 2f were used to analyze iMO recruitment to the (**f**) brain, totaling: $n = 13$ mock, $n = 6$ at 3dpi, $n = 13$ at 5dpi and $n = 12$ at 7dpi/clinical. For **g** HSV-1 infection, the same mice described in Fig. 2g were analyzed for iMO recruitment to the brain. For **e–g**, data are presented in box-whisker plots with individual data points represented by each symbol within the plot for each mouse genotype. The plots are color-coded such that magenta represents data from *Ccr2*$^{−/−}$ RFP, cyan from *Ccr7*$^{−/−}$, black/gray from HET and orange from DKO mice. For **e** LACV-infected and **g** HSV-1 infected mice, a two-way ANOVA mixed-effects analysis with a Geisser-Greenhouse correction was used to compare the percentage of iMOs in the brain of ($F_{(1.005,39.20)} = 12.64$, $p < 0.0010$) LACV and ($F_{(1.496,36.64)} = 5.215$, $p = 0.0167$) HSV-1 mice from mock to 5dpi with a Tukey's multiple-comparison test to compare four groups over multiple time points with an alpha of 0.05. Specific statistics can be found in Table 2. \*, #, % or & symbols associated with a data point indicate the *p*-value for the statistically comparison between that data point and the time point-associated *Ccr2*$^{−/−}$ RFP, *Ccr7*$^{−/−}$, HET or DKO data point respectively is below 0.05. Because not all mice developing clinical disease, a non-parametric Kruskal–Wallis test was used to compare the percentage of iMOs in the brain (**f**, Kruskal–Wallis statistic = 23.63, $p < 0.0001$) of *Ccr2*$^{−/−}$ RFP, *Ccr7*$^{−/−}$, HET and DKO mice at the 7dpi/clinical time point with a Dunn's multiple comparisons test with an alpha of 0.05 to compare specific groups. *P*-values of comparisons to DKO mice are shown. All other statistics can be found in Table 2. Error bars represent SE for all groups. Dpi = days post infection.

effect of iMOs on the development of neurological disease would be missed. To address this, single knockout *Ccr2*$^{−/−}$ RFP and *Ccr7*$^{−/−}$ mice, along with HET and DKO mice were infected with a ~ LD50, $10^2$ PFU dose of LACV. Again, all strains developed disease with similar timing and frequency with *Ccr7*$^{−/−}$ mice developing disease slightly less, and DKO slightly more often (Fig. 4b). No statistical difference was observed between these groups suggesting iMOs play a negligible role in LACV-induced neuropathogenesis.

### LACV-recruited iMOs have higher proinflammatory, proapoptotic and less promitotic transcript expression than HSV-1-recruited iMOs

LACV induces a large influx of iMOs to the brain, relative to that seen with HSV-1 (Fig. 3f vs g). To examine if iMOs from LACV-infected mice were functionally different than the ones recruited during HSV-1 infection, iMOs from the brains of lethally infected LACV and HSV-1 mice were FACs isolated at the clinical time point and their transcriptomes compared via RNAseq analysis (Figs. 5 and 6). To validate the iMO isolation, expression of known iMO (Fig. 5a) and non-iMO (Fig. 5b) transcripts were compared across all samples. The known iMO transcripts were consistently highly expressed in both iMO populations (Fig. 5a), while non-iMO related gene transcripts were relatively low, indicating the iMOs from each virus infection were highly enriched and highly similar for iMO markers.

Direct comparison of gene transcripts showed substantial differences between iMOs from LACV and HSV-1-infected mice (Fig. 5c–f). Transcript expression of cytokines, chemokines and interferon stimulated genes known to be upregulated during viral encephalitis were consistently higher in iMOs from LACV-infected mice compared to HSV-1-infected mice, while type I IFN was decreased (Fig. 5d). Concordantly, proapoptotic factor transcripts including *Cycs*, *Apaf1*, *Fas*, *FasL* and *Casp7* were increased in LACV iMOs while the antiapoptotic *Bcl2* was decreased (Fig. 5c, e)

disease. Interestingly, HET and DKO mice developed neurological disease with similar timing and frequency, suggesting that iMO recruitment to the brain during LACV infection did not significantly influence viral pathogenesis (Fig. 4a).

Possibly, iMOs recruited with a lethal dose of LACV virus may not influence disease because the high virus dose is overwhelming, and a subtle

suggesting enhanced cell death. Surprisingly, other proapoptotic factors including *Casp3*, *Casp6* and *Bax* were largely unchanged between groups indicating only select cell death pathways may be activated during viral encephalitis.

Correlative to the proapoptotic phenotype, transcripts for promitotic factors were largely decreased in LACV-recruited iMOs (Fig. 5c, f) which may indicate impaired immune cell development. Monocyte-derived macrophages have been shown to self-renew at sites of infection[33] which may be critical for controlling infection in the brain. Interestingly, expression of *Ifna2* and *Ifnb1* transcripts was also decreased in LACV iMOs (Fig. 5c, d) which could be an indication of negative feedback inhibition[34] as the type I interferon response has clearly been initiated as evidenced by high interferon-stimulated gene expression. Likewise, expression of the antimicrobial *Mpo* transcript was decreased in LACV monocytes which could indicate impaired phagolysosomal function[35].

## LACV-recruited iMOs are phagocytic, but have decreased transcription of phagolysosomal machinery

The decreased expression of *Mpo* transcript in LACV iMOs could suggest that phagocytosis or other downstream antiviral effector functions may be impaired in LACV-recruited iMOs in the brain. To address this, iMOs collected from the blood and brain of LACV infected mice were evaluated for their phagocytic ability compared to iMOs from HSV-1 infected mice (Fig. 6a–d). FACS isolated iMOs from blood (Fig. 3b, c) or brain (Fig. 3d, e) of clinical mice infected with either HSV-1 or LACV were incubated with fluorescent bioparticles to measure phagocytic uptake and fluorescent intensity was compared to cells incubated without bioparticles (Fig. 3b, d, gray line). The majority of iMOs isolated from both HSV-1 or LACV infected mice were capable of phagocytosing bioparticles indicating iMOs from LACV could be functional effectors. In support of this finding, *Fcgr1* and *Lamp2* transcripts, which are important for phagosome formation[36], were increased in LACV iMOs (Fig. 6e). However, the ability to phagocytose does not confirm function. In fact, expression of multiple other transcripts involved in phagosome formation, function and fusion to the lysosome were decreased in LACV iMOs relative to HSV-1 (Fig. 6e–g). These included the cell surface receptors *Fcgr2b*, *Clec7a*, *P2rx7*, *Manba*, *Lrp1* and *CD36* which can initiate phagocytosis[37,38], as well as multiple intracellular effectors of phagosome internalization and maturation including well known effectors such as *Rab37*, *Atg5*, *Pikfyve* and *Eea1*[39–41]. Transcripts of critical effectors that mediate the fusion of the phagosome with the lysosome into the degradative phagolysosome were also largely decreased in LACV iMOs relative to HSV-1, including components of the HOPS (Fig. 6f) and SNARE (Fig. 6g) complex which facilitate fusion of the two organelles[42]. Thus, these data suggest that despite being functionally phagocytic, LACV-recruited iMOs have impaired transcription of critical factors involved in the formation of the phagosome and phagolysosome which may contribute for their inability to influence disease as has been shown for HSV-1-recruited iMOs.

## Discussion

iMO monocytes have been shown to influence encephalitic disease[11,12,14] and represent a potential therapeutic target[16], but how they are recruited from the bone marrow to the brain is not fully understood. Here, we identified CCR7 as an important receptor for iMO recruitment from the bone marrow to the brain following virus infection. iMO recruitment was reduced, but was still robust, in both CCR2 and CCR7 single KO mice following LACV infection (Fig. 2). However, the absence of both receptors nearly eliminated iMO recruitment to the blood or brain (Figs. 2 and 3). Thus, iMO recruitment from the bone marrow to the brain during LACV infection can be mediated by either CCR2 or CCR7, with an additive effect of these receptors to induce substantial recruitment of iMOs to the brain.

CCR2 has been shown to be important for iMO recruitment during homeostatic and pathologic states for nearly two decades[43]. Many studies, including our own work with the encephalitic viruses HSV-1 and TAHV, have validated this finding[7,11,12,14,22,32,43–45]. However, the current study

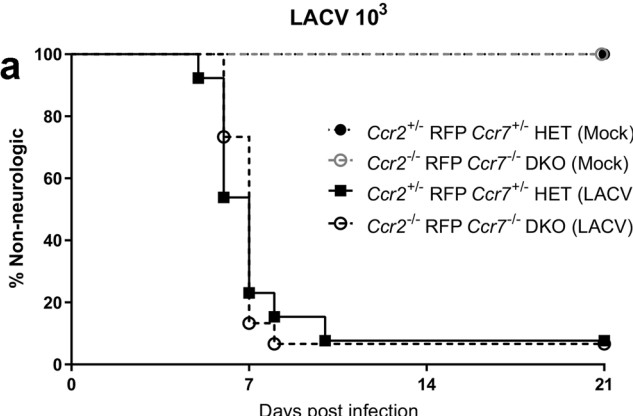

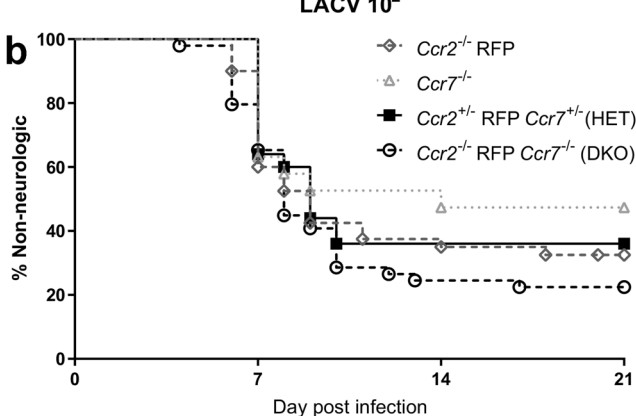

**Fig. 4 | Impaired iMO recruitment does not alter LACV disease. a** $n = 3$ HET and $n = 3$ DKO mice injected with mock inoculum and $n = 13$ HET and $n = 15$ DKO mice infected IP with high dose ($10^3$ PFU) LACV were followed for the development of clinical signs of neurological disease. A Log-rank Mantel–Cox curve comparison test was used to examine differences in survival between HET and DKO animals but did not reach significance (chi Square value = 0.1148, df = 1, $p = 0.7347$). **b** $n = 30$ *Ccr2*$^{-/-}$ RFP, $n = 19$ *Ccr7*$^{-/-}$, $n = 25$ HET and $n = 38$ DKO mice infected IP with a low dose ($10^2$ PFU) of LACV were followed for the development of clinical signs of neurological disease. A Log-rank Mantel-Cox curve comparison test was used to examine differences in survival between *Ccr2*$^{-/-}$ RFP, *Ccr7*$^{-/-}$, HET and DKO mice but did not reach significance (chi Square value = 4.221, df = 3, $p = 0.2386$).

uncovered a previously unappreciated role for CCR7 in iMO recruitment to the blood and brain during encephalitic viral infection and provides a more complete view of how CCR2 and CCR7 have complementary roles in this process under homeostatic conditions. In uninfected *Ccr2*$^{-/-}$ RFP mice, iMOs account for ~1.0–2.5% of CD45$^+$ cells in the blood while they account for ~2.8–4.8% of cells in *Ccr7*$^{-/-}$ mice (Fig. 2e). These amounts summate closely to the 3.9–6.9% of iMOs in blood found in HET controls (Fig. 2e)[7] demonstrating the receptors are complementary even in the absence of an inflammatory signal. Furthermore, iMOs are found at lower levels in DKO mice compared to *Ccr2*$^{-/-}$ RFP or *Ccr7*$^{-/-}$ in mock or infected mice (Fig. 2e). Thus, our data demonstrate that CCR2 and CCR7 cooperate to contribute to the release of iMOs to the blood both during health and disease.

CCR7 expression by monocytes and monocyte-derived dendritic cells has been associated with their migration to the draining lymph node or target tissue in response to inflammation[46–48] or infection[30]. These studies have focused on CCR7 function in monocyte-lineage cells following maturation or after arriving in target tissues. Our current findings suggest that CCR7 also functions upstream of trafficking to work in tandem with CCR2 to promote the initial recruitment of iMOs out of the bone marrow. Little is known about the expression of CCR7 by monocytes in the bone marrow, but transcriptional evidence suggests the receptor is lowly

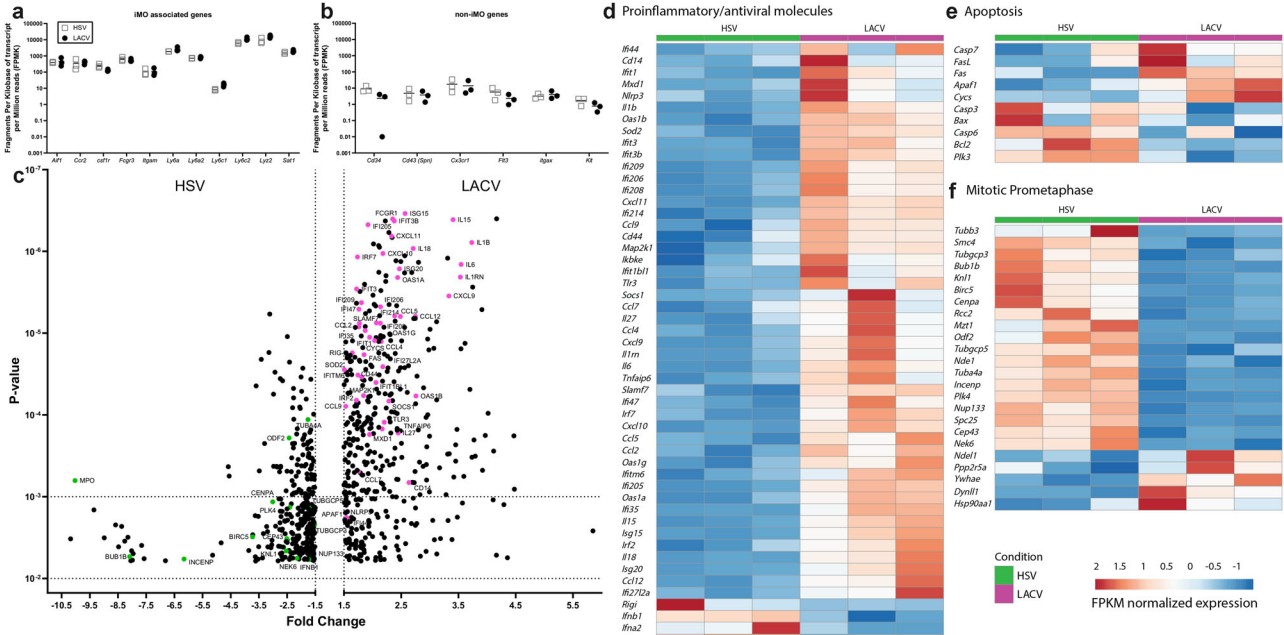

**Fig. 5 | LACV-recruited iMOs express more proinflammatory and proapoptotic, and fewer promitotic transcripts than HSV-1-recruited iMOs.** iMOs from (square) 3 HSV-1- and (circle) 3 LACV-infected brains were assayed for (**a**) iMO associated or (**b**) non-iMO associated transcripts. Data are plotted as fragments per kilobase of transcript per million reads (FPKM). **c** Volcano plot of the top 436 up-regulated and 334 down-regulated transcripts expressed by FACs-isolated iMOs in the brains of LACV and HSV-1 infected mice. Data are plotted as the expression fold change of LACV vs HSV-1 iMO transcripts (x-axis) relative to *p*-value as determined by a Student's *t* test between groups assuming equal variances (y-axis). Heatmaps showing FPMK normalized expression of known (**d**) proinflammatory and antiviral, (**e**) proapoptotic and (**f**) mitotic prometaphase transcripts in iMOs from the brains of 3 LACV and 3 HSV-1 infected mice. Replicate mice from each infection group indicated (HSV-1 = green, LACV = magenta) had increased or decreased expression as shown by red or blue colored boxes respectively. Transcripts from the (**d**, **e** and **f**) heatmaps that were also in the top differentially expressed genes shown in **c** are highlighted in the volcano plot as either (magenta) up- or (green) down-regulated in LACV relative to HSV-1 iMOs.

expressed under normal conditions[49]. However, expression on monocytes substantially increases as they reach the blood[26] suggesting it could be involved in the iMO recruitment process. Furthermore, CCR7 expression cannot be induced following maturation of monocytes into macrophages[50] indicating CCR7 expression is associated with the recruitment and migration but not subsequent maturation of iMOs into effector cells.

The involvement of both CCR2 and CCR7 in the recruitment of iMOs could be an advantage to the immune response in multiple ways. First, redundancy in the signaling response controlling iMO recruitment is beneficial as it limits the ability of pathogens to evolve evasion strategies to escape the immune response[51]. Additionally, the involvement of multiple receptors in this process allows for the response to be tunable, such that a stronger response can be elicited by a weak signal because an overall larger number of receptors could respond to the infection[52]. Conversely, the fact that only CCR7 is required to drive a robust iMO response during a lethal LACV infection[7] indicates that the system can be saturated which would protect the host from an overwhelmingly harmful immune response. This is advantageous as the brain is a very delicate tissue and iMOs can be damaging in certain contexts[15,16]. Finally, it may be beneficial from an evolutionary perspective to have redundancy in immune response signaling that is not centered around a single genomic locus. A recent study demonstrated that deletion of the genomic locus containing the genes for *Ccr1*, *Ccr2*, *Ccr3* and *Ccr5* on chromosome 9 had a minimal impact on monocyte recruitment beyond the phenotype observed by *Ccr2* knockout alone[44] leading the authors to conclude CCR2 is the primary controller of recruitment. However, our data indicate that CCR7, which is located on chromosome 11, plays a complementary role in recruitment that could account for the CCR2-independent monocytes observed by Dyer et al.[44] in *Ccr2* and genome locus knockout tissues during resting and inflamed states.

We observed that iMOs play little-to-no role in LACV-induced pathogenesis regardless of infectious dose administered (Fig. 4). This was a surprising result considering these cells influence the pathogenesis of multiple other encephalitic viruses, including HSV-1[11,12,14,16]. Furthermore, iMOs enter the brain of LACV-infected animals in exceptionally high numbers compared to other encephalitic viruses (Fig. 3f vs g)[7] and are functionally phagocytic (Fig. 6a–d). The limited effect of LACV-recruited iMOs may be explained by their substantially different transcriptomics profile (Figs. 5 and 6), which had higher expression of proinflammatory and proapoptotic transcripts. This heavily proinflammatory/proapoptotic phenotype could be the result of an overwhelming lethal viral infection in neurons[53–55] that is beyond the ability of infiltrating immune cells to control. LACV-recruited iMOs also had lower expression of transcripts associated with type I IFN production, mitosis, phagocytosis and phagolysosomal fusion, which are likely indicative of an immature, ineffective immune phenotype. Thus, infiltration of large numbers of iMOs into the LACV infected brain, particularly ones that may not be fully mature or possibly overstimulated, may simply be insufficient to combat an exceptionally aggressive encephalitic virus. It is possible, that the strong signals driving robust iMO recruitment during LACV infection could lead to this impaired state.

## Methods
### Ethics statement
Mouse experiments were approved by the Rocky Mountain Laboratories (RML) Animal Care and Use Committee and adhered to the National Institutes of Health guidelines and ethical policies. The RML facility is fully accredited by AAALAC International, and we have complied with all relevant ethical regulations for animal use.

### Virus stocks and plaque assay
LACV (human, 1978)[6] and McKrae HSV-1[56] stocks were made using Vero (CCL-81, ATCC) cells by infecting a confluent T75 flask (Corning) with a

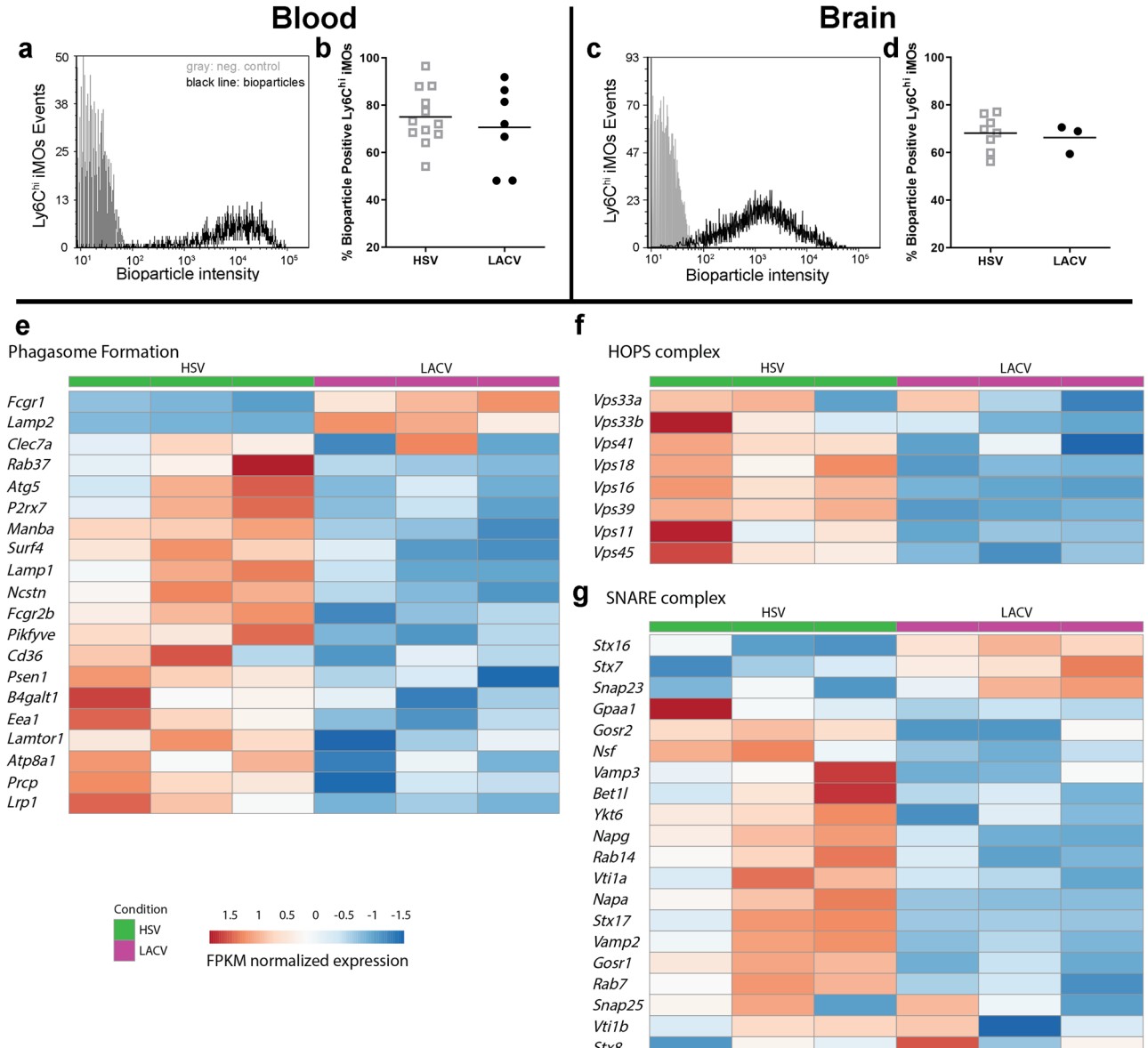

**Fig. 6 | LACV-recruited iMOs are functionally phagocytic but may have impaired phagolysosome fusion.** Flow cytometrically identified iMOs from LACV and HSV-1 infected mouse (**a**, **b**) blood or (**c**, **d**) brain were assayed for their ability to phagocytose labeled bioparticles. Representative histogram plots of negative control (gray plots) and bioparticle-fed (black plots) samples from (**a**) blood and (**c**) brain demonstrate the positive bioparticle signal intensity in iMOs taken from LACV and HSV-1 infected brain. The percent of positive cells in (**b**) blood and (**d**) brain are plotted for each mouse with the black horizontal bar representing the mean. A two-

tailed, unpaired t-tests was performed to examine differences in iMO phagocytosis collected from blood (**c**, t = 0.6497 df = 17, $p$ = 0.5246) or brain (**e**, t = 0.3862, df = 9, $p$ = 0.7083) of LACV and HSV-1 infected mice, that did not reach significance. Heatmaps showing FPMK normalized expression of known (**e**) phagosome-, (**f** and **g**) phagolysosomal-forming transcripts in iMOs from the brain of 3 LACV and 3 HSV-1 infected mice. Replicate mice from each infection group indicated (HSV-1 = green, LACV = magenta) and increased or decreased expression are shown by red or blue colored boxes, respectively.

---

multiplicity of infection of 0.01 of either virus and harvesting the supernatant of each flask 5 days later. Supernatants were titered using plaque assay[55] for LACV or focus forming assay for HSV-1 to determine viral concentrations. For both plaque and focus forming assays, Vero cells were plated 1 day in advance into 24-well plates (Corning) to confluency. Viral stock supernatants were diluted serially at 10-fold decreasing concentrations in DMEM (Gibco) supplemented with 2% FBS and penicillin/streptomycin and 200 μl of each dilution was applied the Vero cultures in duplicate. Plates were incubated 1 h to allow viral attachment and then each well was over-layed with 0.5 mL of 1.5% carboxymethyl cellulose in MEM (Gibco). Plates were incubated for 5 days and then fixed with 10% formaldehyde to a final concentration ≥4% formaldehyde per well for 1 h. HSV-1 wells were rinsed and refilled with 1xPBS to count endogenous viral-GFP foci for each sample

dilution. LACV plates were rinsed with water stained with 0.35% crystal violet and rinsed with water again. LACV plates were air dried, and plaques were counted for each sample dilution. Sample titers were calculated as focus (HSV-1) or plaque (LACV)-forming units (FFU or PFU) per mL of supernatant.

### Mice, husbandry and inoculations

$Ccr2^{-/-}$ RFP (B6.129(Cg)-$Ccr2^{tm2.1lfc}$/J)[32,57], $Ccr7^{-/-}$ (B6.129P2(C)-$Ccr7^{tm1Rfor}$/J)[58] and C57BL/6J (B6) mice were purchased from The Jackson Laboratory and maintained in RML breeding colonies. $Ccr2^{-/-}$ RFP $Ccr7^{-/-}$ double knockout (DKO) mice were generated in-house by crossing single knockouts and selective breeding of genotyped progeny. Confirmation of DKO mice is shown in the relevant figure and the primers used are reported

**Article**

**Table 1 | Statistics associated with analysis of iMOs in the blood of LACV and HSV infected mice in Fig. 2**

| Infection tissue analyzed (figure) | Genotype comparison | Days post infection (dpi) | q-(mixed-effects) or Z-(Kruskal–Wallis) statistic | Degrees of freedom (df, mixed-effects) or mean rank difference (Kruskal–Wallis) | p-value |
|---|---|---|---|---|---|
| LACV 10³ Blood (Fig. 2e) | Ccr2⁻/⁻ RFP vs Ccr7⁻/⁻ | Mock | 4.313 | 4 | 0.1189 |
| | | 3 | 0.8098 | 5 | 0.9361 |
| | | 5 | 0.1277 | 4 | 0.9997 |
| | Ccr2⁻/⁻ RFP vs HET | Mock | 5.666 | 5 | 0.0368 |
| | | 3 | 7.774 | 5 | 0.0102 |
| | | 5 | 6.442 | 5 | 0.0222 |
| | Ccr2⁻/⁻ RFP vs DKO | Mock | 9.687 | 5 | 0.0039 |
| | | 3 | 21.16 | 7 | <0.0001 |
| | | 5 | 13.71 | 6 | 0.0003 |
| | Ccr7⁻/⁻ vs HET | Mock | 2.905 | 4 | 0.3055 |
| | | 3 | 4.015 | 5 | 0.1207 |
| | | 5 | 6.101 | 4 | 0.0414 |
| | Ccr7⁻/⁻ vs DKO | Mock | 8.344 | 4 | 0.0141 |
| | | 3 | 8.657 | 5 | 0.0064 |
| | | 5 | 26.15 | 4 | 0.0002 |
| | HET vs DKO | Mock | 9.840 | 7 | 0.0010 |
| | | 3 | 19.77 | 5 | 0.0001 |
| | | 5 | 21.38 | 5 | <0.0001 |
| LACV 10³ Blood (Fig. 2f) | Ccr2⁻/⁻ RFP vs Ccr7⁻/⁻ | 7 | 1.619 | −8.795 | 0.6325 |
| | Ccr2⁻/⁻ RFP vs HET | 7 | 1.805 | −10.71 | 0.4260 |
| | Ccr2⁻/⁻ RFP vs DKO | 7 | 2.483 | 11.52 | 0.0782 |
| | Ccr7⁻/⁻ vs HET | 7 | 0.3036 | −1.917 | >0.9999 |
| | Ccr7⁻/⁻ vs DKO | 7 | 3.970 | 20.32 | 0.0004 |
| | HET vs DKO | 7 | 3.937 | 22.23 | 0.0005 |
| HSV-1 10⁷ Blood (Fig. 2g) | Ccr2⁻/⁻ RFP vs Ccr7⁻/⁻ | Mock | 8.862 | 7.383 | 0.0015 |
| | | 3 | 2.528 | 4.734 | 0.3802 |
| | | 5/clinical | 0.3282 | 8.998 | 0.9953 |
| | Ccr2⁻/⁻ RFP vs HET | Mock | 10.05 | 5.029 | 0.0032 |
| | | 3 | 11.09 | 6.136 | 0.0008 |
| | | 5/clinical | 4.114 | 5.132 | 0.1095 |
| | Ccr2⁻/⁻ RFP vs DKO | Mock | 7.212 | 7.996 | 0.0041 |
| | | 3 | 8.438 | 12.79 | 0.0003 |
| | | 5/clinical | 6.234 | 5.653 | 0.0199 |
| | Ccr7⁻/⁻ vs HET | Mock | 4.779 | 5.796 | 0.0573 |
| | | 3 | 4.028 | 6.460 | 0.0973 |
| | | 5/clinical | 4.317 | 4.922 | 0.0975 |
| | Ccr7⁻/⁻ vs DKO | Mock | 14.89 | 7.464 | <0.0001 |
| | | 3 | 5.608 | 4.353 | 0.0477 |
| | | 5/clinical | 6.353 | 4.639 | 0.0273 |
| | HET vs DKO | Mock | 13.53 | 5.075 | 0.0008 |
| | | 3 | 16.70 | 5.039 | 0.0003 |
| | | 5/clinical | 6.787 | 4.074 | 0.0279 |

in Supplementary Table 1. Heterozygous $Ccr2^{+/-}$ RFP $Ccr7^{+/-}$ (HET) controls were generated by crossing DKO mice with B6 mice. Mice were co-housed with same-sex littermates in ventilated plastic cages (Innovive IVC Caging Systems) with autoclaved bedding (Sani-Chips®) and enrichment (Shepherd Specialty Papers). Food and water were provided *ad libitum*. Animal holding rooms were kept at $22 \pm 2\,°C$ with $50 \pm 10\%$ relative humidity, and a 12-h light-dark cycle. All experimental groups were mixed-sex, except the RT² Profiler PCR Array where only male mice were used to

minimize sex-specific transcriptional hits per the manufacturer's recommendations. Experimental mouse numbers are indicated in the associated figure legend.

Animals were reared in a specific pathogen free facility maintained by biannual sentinel sampling. Experimental animals were housed in an approved animal biosafety level 2 (ABSL-2) vivarium. Virus inoculations were performed intraperitoneally (IP) in weanling mice between the age of 20–22 days old. Virus stocks were diluted in phosphate-buffered saline

**Table 2 | Statistics associated with analysis of iMOs brain of LACV and HSV-1 infected mice in Fig. 3**

| Infection tissue analyzed (figure) | Genotype comparison | Days post infection (dpi) | q-(mixed-effects) or Z-(Kruskal–Wallis) statistic | Degrees of freedom (df, Mixed-effects) or mean rank difference (Kruskal–Wallis) | p-value |
|---|---|---|---|---|---|
| LACV $10^3$ Brain (Fig. 3e) | $Ccr2^{-/-}$ RFP vs $Ccr7^{-/-}$ | Mock | 3.158 | 5.139 | 0.2308 |
| | | 3 | 1.259 | 8.612 | 0.8103 |
| | | 5 | 1.176 | 4.118 | 0.8381 |
| | $Ccr2^{-/-}$ RFP vs HET | Mock | 2.856 | 10.31 | 0.2421 |
| | | 3 | 6.498 | 7.147 | 0.0099 |
| | | 5 | 7.255 | 10.95 | 0.0016 |
| | $Ccr2^{-/-}$ RFP vs DKO | Mock | 3.084 | 9.083 | 0.1995 |
| | | 3 | 6.326 | 7.085 | 0.0116 |
| | | 5 | 2.141 | 6.340 | 0.4824 |
| | $Ccr7^{-/-}$ vs HET | Mock | 4.552 | 4.720 | 0.0858 |
| | | 3 | 5.480 | 5.040 | 0.0412 |
| | | 5 | 0.02722 | 4.086 | >0.9999 |
| | $Ccr7^{-/-}$ vs DKO | Mock | 4.622 | 4.453 | 0.0870 |
| | | 3 | 5.370 | 5.023 | 0.0447 |
| | | 5 | 1.447 | 4.003 | 0.7470 |
| | HET vs DKO | Mock | 0.001687 | 15.07 | >0.9999 |
| | | 3 | 1.450 | 9.338 | 0.7395 |
| | | 5 | 13.46 | 5.390 | 0.0006 |
| LACV $10^3$ Brain (Fig. 3f) | $Ccr2^{-/-}$ RFP vs $Ccr7^{-/-}$ | 7 | 0.8676 | −4.364 | >0.9999 |
| | $Ccr2^{-/-}$ RFP vs HET | 7 | 1.887 | −10.36 | 0.3554 |
| | $Ccr2^{-/-}$ RFP vs DKO | 7 | 2.907 | 13.14 | 0.0219 |
| | $Ccr7^{-/-}$ vs HET | 7 | 1.026 | −6.000 | >0.9999 |
| | $Ccr7^{-/-}$ vs DKO | 7 | 3.542 | 17.50 | 0.0024 |
| | HET vs DKO | 7 | 4.342 | 23.50 | <0.0001 |
| HSV-1 $10^7$ Brain (Fig. 3g) | $Ccr2^{-/-}$ RFP vs $Ccr7^{-/-}$ | Mock | 3.098 | 7.927 | 0.2059 |
| | | 3 | 1.184 | 10.32 | 0.8359 |
| | | 5 | 1.081 | 6.054 | 0.8675 |
| | $Ccr2^{-/-}$ RFP vs HET | Mock | 1.769 | 4.251 | 0.6308 |
| | | 3 | 1.986 | 7.896 | 0.5309 |
| | | 5 | 1.965 | 8.867 | 0.5358 |
| | $Ccr2^{-/-}$ RFP vs DKO | Mock | 1.409 | 4.221 | 0.7597 |
| | | 3 | 0.7270 | 8.334 | 0.9535 |
| | | 5 | 2.588 | 5.109 | 0.3579 |
| | $Ccr7^{-/-}$ vs HET | Mock | 6.464 | 4.304 | 0.0297 |
| | | 3 | 0.3947 | 4.554 | 0.9914 |
| | | 5 | 0.06538 | 5.281 | >0.9999 |
| | $Ccr7^{-/-}$ vs DKO | Mock | 6.081 | 4.268 | 0.0374 |
| | | 3 | 2.361 | 4.2080 | 0.4266 |
| | | 5 | 2.550 | 4.023 | 0.3881 |
| | HET vs DKO | Mock | 1.512 | 7.969 | 0.7165 |
| | | 3 | 6.995 | 11.70 | 0.0018 |
| | | 5 | 6.414 | 4.140 | 0.0329 |

(PBS) to the required concentration in a 200 µl volume for IP administration. Supernatant from uninfected Vero cultures equivalent diluted in 200 µl of PBS and administered via the same route served as a mock control. A lethal dose of LACV was $10^3$ PFU/mouse and an ~50% lethal dose (LD50) was $10^2$ PFU/mouse. HSV-1 was given at $10^7$ FFU/mouse for all experiments.

For the RT2 Profiler PCR array, RNA-seq, flow cytometry, phagocytosis and qRT experiments, the investigator was responsible for mouse infection and harvesting of tissues and was thus unblinded for this process. However, after tissue collection, all experimental animal tissues were assigned a de-identified sequential tracking number which followed the specific tissue through the processing, experimentation, and analysis pipeline. Only after the final analysis was completed were the tracking numbers cross-reference with the experimental groups to unblind the investigator. For survival studies, the investigator was not blinded because they were responsible for randomizing groups by sex and treatment group and

**Table 3 | List of flow cytometry and FACs antibodies used to identify immune cell of interest**

|  | Antigen | Fluorochrome | Clone | Source | Catalog# | Lot# |
|---|---|---|---|---|---|---|
| Flow cytometry antibody panel | CD45 | APC/Cy7 | 30-F11 | BD Biosciences | 557659 | 7215837 |
|  | CD11b | BV510 | M1/70 | BioLegend | 101245 | B360991 |
|  | Ly6C | AF700 | AL-21 | BD Biosciences | 561237 | 0293151 |
|  | Ly6G | Pacific Blue | 1A8 | BioLegend | 127612 | B288476 |
|  | IA/IE | PerCP/Cy5.5 | M5/114.15.2 | BD Biosciences | 562363 | B253463 |
|  | CD80 | APC | 16-10A1 | BioLegend | 104714 | B331465 |
|  | F480 | BV605 | BM8 | BioLegend | 123133 | 4329685 |
|  | CD11c | FITC | HL3 | BD Biosciences | 553801 | 9352193 |
|  | Ly6G | BV395 | 1A8 | BD Biosciences | 563978 | 9343809 |
|  | CCR7 | BV421 | 4B12 | BD Biosciences | 562675 | 1284362 |
|  | cKit | PE/Cy7 | 2B8 | BioLegend | 105814 | B205421 |
| FACs antibody panel | CD45 | PE | 30-F11 | BD Biosciences | 553801 | 2300873 |
|  | CD11b | BV510 | M1/70 | BioLegend | 101245 | B360991 |
|  | Ly6C | AF700 | AL-21 | BD Biosciences | 561237 | 0293151 |
|  | Ly6G | Pacific Blue | 1A8 | BioLegend | 127612 | B288476 |
|  | CD3 | PerCP/Cy5.5 | 17A2 | BD Biosciences | 560591 | B207079 |
|  | CD11c | PE/Cy7 | HL3 | BD Biosciences | 561022 | 7319586 |

performed the injections. However, monitoring animals for development of disease is not subjective and any clinical signs of neurological disease including impaired ambulation, paralysis, ataxia, seizures, or repetitive tics resulted in a clinical score. Clinical disease progresses rapidly in mice infected with LACV and a mouse showing mild neurological signs will progress to severe disease within a matter of hours, thus a clinical score is consistently accurate to that day. Mice were twice monitored daily for presentation of clinical neurological signs.

## Flow cytometry

At indicated timepoints post-infection, brain or whole bone marrow was collected into ice cold PBS, and blood was collected in a 1 mL syringe attached to a 27-gauge 1/2" needle with the needle barrel filled with 1000 U/mL heparin. Whole bone marrow was passed through a 70 μm filter to generate a single cell suspension and was then incubated with LIVE/DEAD fixable blue (ThermoFisher) at 1 μl of working solution per $10^6$ cells for 30 min (min) to identify live cells prior to further processing. Bone marrow cells and immune cells that were isolated from brain and blood were then incubated with CD16/CD32 FcγIII/II (BD Biosciences, clone 2.4G2) to block Fc-receptors and immunolabeled with the primary conjugate antibodies indicated in Table 3, each at a 1:200 dilution for 30 min at 4 °C in the dark[5,7]. RFP fluorescence expression in the place of CCR2 on iMOs was observed in $Ccr2^{-/-}$ RFP, DKO and HET mice. Cells were then fixed with 2% paraformaldehyde for 30 min, washed 3× with PBS and analyzed using a BD LSRII or FACsSymphony A5 (BD Biosciences). The gating strategy for identifying iMO progenitors in bone marrow is shown in Supplementary Fig. 2 and the gating strategy for identifying iMOs in blood is shown in Supplementary Fig. 1. iMO identification in the brain has been published[7] but utilizes the same gating strategy as in blood with the exception that microglia cells must be excluded as is shown in Fig. 1a, c. Antibodies listed in Table 3, but not shown in the gating strategy were used solely for confirmation of negative expression within iMOs and were not part of the core gating strategy. Single-stain and fluorescence minus one controls were performed for all experiments to establish gating boundaries. Data analysis was performed using FCS Expression Research Edition version 5 (Denovo software).

## FACs

B6 mice were infected with either $10^3$ LACV or $10^7$ HSV-1 IP and brain immune cells were isolated as described above. Isolated cells were processed and immunolabeled as described above for identification of iMOs by FACs with either a FACsAria II (BD Biosciences) for PCR analysis[59] or, for RNA-seq analysis, using a Miltenyi MACSQuant Tyto cell sorter (Miltenyi Biotec) per the manufacturer's recommendations. Briefly, for RNA-seq analysis, a MACSQuant Tyto HS cartridge was primed with 0.4 mL of running buffer and brain immune cell samples were loaded into the cartridge. Prior to sorting, a small volume (~50 ul) of each sample were run by Flow Cytometry to define exclusion gates for debris and undesired cell types and to accurately identify iMOs. The antibodies used are shown in Table 3. The gating strategy used for FACs identification of iMOs was the same as for Flow Cytometry and is shown in Supplementary Fig. 1. Antibodies listed in Table 3, but not shown in the gating strategy were used solely for confirmation of negative expression and assessment of sort purity. FACs isolated iMOs were collected directly into 300 ul of RLT buffer for further processing as discussed below.

## RT² profiler PCR array analysis

iMOs were collected from the brain via FACs sorting as described above at the clinical time point. Whole-cell RNA was extracted from iMOs using Zymo RNA isolation kits per the manufacturer's protocol. Extracted RNA was analyzed by a Nanodrop spectrophotometer (Thermofisher) to ensure sufficient concentration and purify to proceed. Using 200 ng of RNA from each sample, genomic DNA elimination, reverse transcription first strand cDNA synthesis and then real-time qRT (q-RT) analysis of the cDNA was performed per the manufacturer's RT² Profiler PCR Array for GPCRs (Qiagen, PAMM-3009ZE) protocol. qRT was performed on a Viia7 thermocycler (Life Technologies). $C_T$ values for 370 genes were analyzed using the RT² Profiler PCR Array data analysis patch. All samples passed internal quality control standards for RT efficiency, specificity and genomic contamination.

## Phagocytosis assay

Prior to tissue collection, *Escherichia coli* (K-12), Alexa Flour 488 conjugated bioparticles (Thermo Fisher, E13231) were reconstituted at 3.3 mg/mL in PBS and sonicated 2 × 15 s at 50 kHz with 65 W. Bioparticles were pre-opsonized by combining 1:1 with mock infected B6 mouse plasma at 37 °C for 30 min and then placed on ice until use. Immune cells from brain and blood were isolated and split between two wells of a 96 well plate (50 μl and 25 μl/well respectively) for each mouse. Each well's volume was brought up to 140 μl

with Hank's buffered salt solution with 5% fetal bovine serum (FBS). 10 μl of bioparticles were added to one well of each mouse tissue and 10 μl of mouse serum mixed 1:1 with PBS was added to the other well to serve as a negative control. The plate was incubated for 30 min at 37°C and then 100 μl of chilled 0.1% trypan blue solution in PBS was added to each well for 1 min to quench any extracellular fluorescence. Cells were pelleted at $500 \times g$ for 3 min, washed 2X in PBS with 2% FBS and prepared for flow cytometry as described above with the exception that the FITC channel was left open to measure the bioparticles.

### RNA isolations and real-time qPCR analysis

RNA was extracted from ½ brains, and inguinal lymph nodes following a Trizol and chloroform protocol[60]. Briefly, brains and lymph nodes were placed in 2 mL tubes (Sarstedt) containing 1.5 and 1 ml of Trizol respectively and 3–5, 2.3 mm Zirconia/Silica beads (Fisher). Tubes were placed on a bead mill for 5200 rpm for $20 \text{ s} \times 2$ with 5 s of dwell to homogenize tissues. Samples were then transferred to 1.5 ml microcentrifuge tubes and 200 ul of chloroform was added. Samples were shaken vigorously, then centrifuged at $12,000 \times g$ for 15 min. The aqueous phase was transferred to a new tube and RNA was precipitated with 600 ul isopropanol for ~15 min–1 h. RNA was then pelleted at $12,000 \times g$ for 10 min and the supernatant was removed. The pellet was washed in 1 ml 70% EtOH, vortexed, and centrifuged at $7600 \times g$ for 5 min. The supernatant was removed and the pellets were allowed to air-dry for ~5 min. Brain and lymph node RNA was resuspended in 100 μl and 50 μl of nuclease-free water respectively and incubated at 55 °C for 10 min. Samples not used immediately were stored at −80 °C. 5 μl of brain and 50 μl of lymph node RNA was then DNase treated using the DNase I kit instructions (Ambion DNase). DNase-treated samples were then cleaned up with the Zymo RNA cleanup kit following the manufacturer's instructions, with the exception that RNA wash steps were performed with 600 ul Wash Buffer and RNA was eluted in 50 μl warm NF H$_2$O. cDNA was synthesized from the cleaned-up RNA using the BioRad iScript cDNA synthesis kit, following the kit instructions. cDNA samples were diluted five-fold in nuclease-free water and quantitative real-time PCR reactions were set up in triplicate in 384-well plates using SYBR Green SuperMix with ROX (Bio-Rad Laboratories) per the manufacturer's recommendation with each primer at a 10 mM concentration. cDNA without reverse transcriptase and nuclease-free water with reverse transcriptase were used as negative controls. All samples were run on a Viia7 (Applied Biosystems) with a 95 °C dissociation for 3 min followed by 40 cycles of 95 °C/15 s to 60 °C/1 min amplification. A final 60–95 °C dissociation melt curve was generated to confirm amplification of a single product for each primer pair in each sample. The percentage difference in C$_T$ values with the housekeeping gene Gapdh were calculated ($\Delta$C$_T$ = (C$_T$ *Gapdh*) − (C$_T$ gene of interest)) for each sample ("%gapdh"). Fold changes of virus and chemokines were calculated comparing the %gapdh values of mock controls with infected samples[61]. The viral and chemokine-specific primers used are shown in Supplementary Table 2.

### RNA-seq processing and analysis

Brain immune cells from 3 HSV-1 and 3 LACV (both 2 females and 1 male) infected mice were prepared and labeled for FACs isolation as described above. Isolated iMOs from were lysed in RLT lysis buffer. The lysate was combined with additional RLT buffer and beta mercaptoethanol (MilliporeSigma, St. Louis, MO) to bring the final volume to 700 μL with 1% BME. Samples were passed through QIAshredder column (Qiagen, Valencia, CA) at $21,000 \times g$ for 2 min to homogenize any genomic DNA. RNA was extracted using Qiagen AllPrep DNA/RNA mini columns (Valencia, CA). RNA integrity was assessed using the Agilent 2100 Bioanalyzer using RNA 6000 Pico kit (Agilent Technologies, Santa Clara, CA). RNA was quantitated using a fluorescence assay (Quant-it RiboGreen RNA, Thermofisher Scientific, Waltham, MA) on a Tecan Spark multi-plate reader (Tecan, Switzerland). Starting with 200 pg of high-quality total RNA for each sample, volumes were concentrated via speed-vacuum to

9.5 ul. Sequencing libraries were constructed following the Takara SMART-Seq v4 PLUS protocol 011820 (Takara, Mountain View, CA) beginning with oligo(dT) priming/cDNA synthesis and following the manufacturers recommendations. Modifications during cDNA amplification included the use of SeqAmp CB PCR Buffer with 14 cycles of PCR. After AmPure XP bead purification (Agencourt Biosciences, Beverly, MA), cDNA quality was visualized and quantity assessed on a BioAnalyzer HS chip (Agilent Technologies, Inc., Santa Clara, CA). Purified cDNA for each sample was normalized to 1.5 ng in 8 ul volume going into library preparation with the Stem-Loop Adaptor mix. Following 15 cycles of library amplification using balanced SMARTer RNA unique-dual index primers from 96U SetA kit 042121 for Illumina sequencing (Takara, Mountain View, CA), individual libraries were cleaned using an 80% vol of AmPure XP beads, visualized on a BioAnalyzer HS chip, and quantified using the Kapa SYBR FAST Universal qPCR kit for Illumina sequencing (Kapa Biosystems, Wilmington, MA) on the CFX384 Real-Time PCR Detection System (Bio-Rad Laboratories, Hercules, CA). Each final library was diluted to a final concentration of 1.5 nM and pooled together in equimolar concentrations for sequencing. After an initial MiSeq paired-end 2 ×150 cycle sequencing run was completed to confirm proper index balancing, samples were run on a NextSeq2k instrument using 1000pM of manually-denatured final library pool and sequenced on a P2 flowcell following a paired-end 2 ×100 run with 300 cycle chemistry. Resulting Raw fastq files were trimmed to remove adapters and low-quality bases using Cutadapt v1.18[62] before alignment to the GRCh39 reference genome and the Gencode v30 genome annotation using STAR v2.7.9a[63]. PCR duplicates were marked using the MarkDuplicates tool from the Picard v3.1.0 (https://broadinstitute.github.io/picard/) software suite. Raw gene counts were generated using RSEM v1.3.3[64] and differential expression was evaluated using DESeq2[65]. Heatmaps for gene sets of interest were generated using Clustvis[66], with FPKM normalized expression as input and rows mean centered.

### Statistics and reproducibility

Sample size was determined by multiple factors. For the RT2 Profiler PCR array and RNA-seq analysis, the number of mice used in each group (3) was dictated respectively by the array and the sequencing chip to reach an adequate sequencing depth of coverage. For flow cytometry, phagocytosis, qRT and survival analyses, an initial power analysis was performed to estimate the number of animals required to detect a specified mean fold change and standard deviation, that were based on results from previous experiments, between groups assuming an alpha of 0.05 and 80% power. These estimates were 6 mice per group for cytometry, phagocytosis and qRT analyses and 14 mice for survival analyses. The final number of mice used in each experimental group was further dictated by the accumulated mean and variance within a group as additional replicates were added. If means and variance were consistent within a group after analyzing 5 or more mice, no additional mice were used. The breeding success of a specific strain, such as *Ccr7*$^{-/-}$ mice also limited the number of mice that were available for analysis. Specific animal numbers are reported in figures and figure legends. All measurements were taken from distinct experimental animals at an indicated time point.

Statistical tests were performed using Prism 9.3.1 software (GraphPad) or, Clustvis written in R (The R Project) for the RNA-seq analysis. As required for the experiment, one of the following statistical tests was used: a two-tailed unpaired t-test, a two-way ANOVA with a Sidak's multiple-comparison or a two-way ANOVA with a Tukey multiple comparison. The specific test(s), degrees of freedom (df) and t-, or F-values and the p-value for each analysis are described in the corresponding figure legend or in the indicated tables. For RNA-seq heatmaps, rows and columns are clustered using Pearson correlation.

### Reporting summary

Further information on research design is available in the Nature Portfolio Reporting Summary linked to this article.

## Data availability

Source data for Fig. 1e are provided in Supplementary Data 1. Numerical source data for all Figures can be found in Supplementary Data 2. RNA-seq analysis data from Figs. 5 and 6 have been upload to NCBI Geo and can be accessed with Accession # GSE254432. All other data that support the findings of this study are available from the corresponding author (CWW) upon reasonable request.

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

## Acknowledgements
We thank Lara Myers, Carrie Long, Lydia Roberts and Christine Schneider-Lewis for critical reading of the manuscript, the entire Rocky Mountain Veterinary Branch staff and especially Jeff Severson, Shelby Heinz, Kimberly Dillard and Rebecca Charlesworth for outstanding animal husbandry with immune compromised animals. We also thank Kishore Kanakabandi, and Kent Barbian from the Research Technologies Branch at Rocky Mountain Laboratories for their assistance processing and running samples for RNA-seq. This work was funded by the Intramural Research Program at NIAID.

## Author contributions
C.W.W. developed study concept, designed, and conducted experiments, wrote the first draft of the manuscript and revised subsequent drafts. A.B.E. designed and conducted experiments. A.B.C. conducted experiments, curated, and analyzed data. J.B.L. analyzed data and generated figures. T.A.W. designed and conducted experiments. K.E.P. developed study concept, reviewed, and revised the manuscript.

## Funding

## Competing interests
The authors declare no competing interests.
