## [Peer review file · Communications Biology]

Reviewers' comments:

Reviewer #1 (Remarks to the Author):

The authors present a manuscript entitled "CCR7 plays a complementary role to CCR2 during monocyte recruitment from the bone marrow". In this article, the authors performed RT² Profiler PCR Array of 370 GPCR transcripts in iMOs isolated from LACV or HSV injected mice. They found Ccr7 transcript is highly expressed in LACV-recruited iMOs. By using Ccr2/Ccr7 double knockout mice, the authors demonstrated that iMOs recruitment depends on CCR2 and CCR7 from bone marrow to blood.

Overall, the manuscript is interesting and in parts convincing, but there are some major issues with each part of this broad manuscript that need to be resolved, especially related with the iMOs in bone marrow.

Points to address:

1. In figure 1e, Ccr7 transcript level was measured but protein level in iMOs is still unknown.
2. In figure 1f, the authors chose 103 pfu LACV for infection and performed analysis at day 7 post-infection. From the survival data that showed in figure 3a, some mice already died at day 7. Day 3 or day 5 may be a better time point to choose for this experiment.
3. From figure 2c and figure 4a, Ly6chi monocytes recruitment to the blood was impaired in Ccr2/Ccr7 DKO mice, so the authors concluded iMOs recruitment from the bone marrow to the blood required CCR2 and CCR7 signals during LACV infection. However, iMOs in bone marrow was never measured, because it is possible that CCR2 and CCR7 signals control iMOs survival in bone marrow and resulted in inhibited recruitment to the blood.
4. To study the role of CCR2/CCR7 controlling iMOs migration from blood to brain, the authors may label bone marrow cells isolated from Ccr2/Ccr7 DKO mice, then intravenously transfer them to WT mice, and measure iMOs level in blood and brain.
5. The gating strategy of FACS is not shown in the manuscript.

Reviewer #2 (Remarks to the Author):

This manuscript by Winkler et al. shows the complementary role of chemokine receptor CCR7 in CCR2-mediated monocyte release from bone marrow. For this purpose, authors used single and double gene KO mice and mouse model of LACV-induced encephalitis to study the monocyte release as well as their role in LACV infection. The authors conclude that while CCR7 plays a complementary role in CCR2-dependent monocyte release, they are dispensable in the control of lethal or sublethal LACV infection. The manuscript is well-written, and the findings are presented in a logical manner. However, there are several concerns, including the lack of proper controls and over-interpretation of data. My comments are following:

1. Fig. 1. The double positive population of CD45 and CD11b seems confusing (1a). Can authors first gate total CD45 population and then gate CD11b before gating Ly6G vs Ly6C. Further, the authors should show the data from both single CCR2 KO and CCR7 KO models under fig. 1f. Even though CCR2 data is published, the relative role of both receptors in Fig 1f would be important to show.
2. Similarly, in Fig. 2, authors should show the data from all four groups, including single KO of CCR2 and CCR7.
3. Fig. 3. Authors conclude that iMOs have no role in the control of lethal (103) and sublethal (102) LACV infection, despite their competent phagocytic response. Phagocytosis alone is not sufficient to draw the conclusions. For the nature of this publication, this reviewer would like to see a comprehensive analysis of monocyte inflammatory responses in all groups of mice (single and double KOs), including ROS production and cytokine/chemokine expression.
4. The data under fig. 4 seem highly variable and hard to draw meaningful conclusions.

5. It seems that the findings are more useful in showing the complementary roles of CCR2 and CCR7 in monocyte egress from bone marrow than the actual control of viral pathogenesis in brain. Therefore, the introduction and discussion sections should be revised to highlight this crucial aspect of their findings.

Reviewer #3 (Remarks to the Author):

The paper "CCR7 plays a complementary role to CCR2 during monocyte recruitment from the bone marrow" by Winkler et al investigate the roles of both CCR2 and CCR7 in iMO recruitment to the brain in LACV versus HSV infection and qualified the roles for both receptors. The experiments use various mouse strains and are carefully constructed. The methods used are sound and the results are novel. Negative points are: The manuscript should be edited for better understanding. Introduction and abstract could be clearer. The two ko strains used were sourced from Jackson but the original publications would be helpful to know.

Reviewer #1 (Remarks to the Author):

The authors present a manuscript entitled "CCR7 plays a complementary role to CCR2 during monocyte recruitment from the bone marrow". In this article, the authors performed RT² Profiler PCR Array of 370 GPCR transcripts in iMOs isolated from LACV or HSV injected mice. They found *Ccr7* transcript is highly expressed in LACV-recruited iMOs. By using *Ccr2/Ccr7* double knockout mice, the authors demonstrated that iMOs recruitment depends on CCR2 and CCR7 from bone marrow to blood. Overall, the manuscript is interesting and in parts convincing, but there are some major issues with each part of this broad manuscript that need to be resolved, especially related with the iMOs in bone marrow.

Points to address:

1. In figure 1e, **Ccr7** transcript level was measured but **protein level** in iMOs is still unknown.

We agree with the reviewer that it is important to confirm the increased CCR7 expression on iMOs recruited during LACV infection relative to HSV. Therefore, we measured CCR7 expression via mean fluorescent intensity (MFI) using flow cytometry in iMOs recruited to the brains of LACV and HSV infected mice which confirmed our transcriptomic finding. These data are shown in current Figure 1 f and g.

2. In figure 1f, the authors chose 103 pfu LACV for infection and performed analysis at day 7 post-infection. From the survival data that showed in figure 3a, some mice already died at day 7. Day 3 or day 5 may be a better time point to choose for this experiment.

The reviewer is correct that some mice do die by 7 post infection. However, this day was chosen for analysis because it is the peak of iMO recruitment to the brain during LACV infection (Winkler et al, 2018 //) and thus the best time to observe differences in recruitment. Furthermore, this time point is the same as was used for the RT2 Prolifer data experiment and we desired to remain consistent. We have clarified the rationale for choosing this time point in the revised manuscript (lines 633-634).

3. From figure 2c and figure 4a, Ly6chi monocytes recruitment to the blood was impaired in *Ccr2/Ccr7* DKO mice, so the authors concluded iMOs recruitment from the bone marrow to the blood required CCR2 and CCR7 signals during LACV infection. However, iMOs in bone marrow was never measured, because it is possible that CCR2 and CCR7 signals control iMOs survival in bone marrow and resulted in inhibited recruitment to the blood.

We thank the reviewer for bringing up this intriguing idea. To address this point, we analyzed the bone marrow from HET and DKO mice for the proportion of live vs. dead iMO progenitors. We found that between groups, the proportion of live iMO progenitors was ~95% or higher regardless of if mice were mock, LACV or HSV infected indicating receptor knockout does not impact progenitor survival. These data are shown in current Supplementary Figure 2.

4. To study the role of CCR2/CCR7 controlling iMOs migration from blood to brain, the authors may label

bone marrow cells isolated from Ccr2/Ccr7 DKO mice, then intravenously transfer them to WT mice, and measure iMOs level in blood and brain.

Our current data show that dual depletion of CCR2 and CCR7 dramatically inhibits iMO recruitment from the bone marrow to the blood (current Figure 2) subsequently largely eliminating iMO recruitment to the final target, the brain (current Figure 3). Although we agree with the reviewer that cell transfer studies of knockout cells into WT recipients could be useful to specifically measure iMO recruitment from the blood to the brain, our goal was to examine the role of CCR2 and CCR7 throughout iMO recruitment. The suggested experiment would bypass the initial step in the recruitment process, so it could not be used to address our primary question.

5. The gating strategy of FACS is not shown in the manuscript.

In addition to the gating strategy shown in Figure 1a-d, we have clarified that the gating strategy shown in updated Supplementary Figure 1b-m was used for all flow and FACS experiments. Additional gates and arrows were also added to Supplementary Figure 1b-m to indicate specific cell populations and their movement through the strategy. We have also clarified that all other antibodies not shown in the gating strategy were used for negative expression to confirm iMOs identity and not part of the core gating strategy (current lines 351-352 and 366-367).

Reviewer #2 (Remarks to the Author):

This manuscript by Winkler et al. show the complementary role of chemokine receptor CCR7 in CCR2-mediated monocyte release from bone marrow. For this purpose, authors used single and double gene KO mice and mouse model of LACV-induced encephalitis to study the monocyte release as well as their role in LACV infection. The authors conclude that while CCR7 plays a complementary role in CCR2-dependent monocyte release, they are dispensable in the control of lethal or sublethal LACV infection. The manuscript is well-written, and the findings are presented in a logical manner. However, there are several concerns, including the lack of proper controls and over-interpretation of data. My comments are following:

1. **Point A.** Fig. 1. The double positive population of CD45 and CD11b seems confusing (1a). Can authors first gate total CD45 population and then gate CD11b before gating Ly6G vs Ly6C.

Point B. Further, the authors should show the data from both single CCR2 KO and CCR7 KO models under fig. 1f. Even though CCR2 data is published, the relative role of both receptors in Fig 1f would be important to show.

Regarding point A, CD45 by CD11b dual gating is the standard for CNS analysis to distinguish microglia versus infiltrating myeloid cells. However, we agree with the review that identifying specific cell populations can be confusing. Therefore, we have color-coded the cell types in Figure 1a-d to assist in identification of the referenced cell types. In Figure 1a and c, microglia are shown as gray to differentiate them from the gated infiltrating CD45⁺ immune cells (contained in the black gate). Within those infiltrating immune cells, taken through to Figure 1b and d, granulocyte/neutrophils are shown as blue and iMOs are

shown as red. The red and blue colors are also overlaid on Figure 1a and c to show the CD11b expression of neutrophils and iMOs. These descriptions were added to the Figure legend.

Regarding point B, we agree with the reviewer examining iMO infiltration into the brains of single CCR2 and CCR7 KO mice is important to show. Thus, we conducted additional experimentation to look at this process throughout infection in both the blood (current Figure 2e and f) and brain (current Figure 3e and f) of single CCR2 and CCR7 KO mice. These new data have been analyzed along with the existing iMO recruitment data that already existed for HET and DKO animals. We have retained the original CCR7 single knockout iMO recruitment data to the brain (current Figure 1h) because this was our initial characterization of these knockouts leading to our generation of the DKO mice and the comparison was done using WT controls rather than the HET mice used in later experiments.

2. Similarly, in Fig. 2, authors should show the data from all four groups, including single KO of CCR2 and CCR7.

As stated in the previous response, we agree with the reviewer and have performed additional experimentation to address this point. This new data is shown in Figure 2e and f and Figure 3e and f.

3. Fig. 3. Authors conclude that IMs have no role in the control of lethal (103) and sublethal (102) LACV infection, despite their competent phagocytic response. Phagocytosis alone is not sufficient to draw the conclusions. For the nature of this publication, this reviewer would like to see a comprehensive analysis of monocyte inflammatory responses in all groups of mice (single and double KOs), including ROS production and cytokine/chemokine expression.

We agree that measuring phagocytosis alone does not demonstrate that iMOs are functional and have clarified this in the revised manuscript (current lines 235-236 and 303-305). However, a direct comparison of all groups is not feasible because in the critical DKO group, the proportions of recruited iMOs is exceedingly low in both blood (current Figure 2) and brain (current Figure 3).

Thus, we devised an alternate experiment to further our analysis and to attempt to understand lack of iMO involvement in LACV disease. We performed an RNA-seq comparison of iMOs from HSV infected brain, which are known to ameliorate disease, and iMOs from LACV infected brain, which do not appear to influence disease (current Figure 4). These new data (shown in current Figure 5 and 6) focus not only on the monocyte inflammatory response as requested, but also on iMO survival, maturation, and phagosome/phagolysosome formation. Our findings indicate that LACV-recruited iMOs have higher expression of proinflammatory transcripts, including interferon stimulated genes, but lower expression of interferon alpha and beta which could indicate an exhausted, ineffective immune response. Additionally, LACV iMOs have higher apoptotic and lower mitotic transcripts indicating the population is unstable. Finally, LACV iMOs have generally lower expression of phagocytic, phagolysosomal and ROS transcripts suggesting they may be functionally impaired.

4. The data under fig. 4 seem highly variable and hard to draw meaningful conclusions.

We agree with the reviewer that the data in the old Figure 4a was highly variable which was likely due to the low dose of virus used in those experiments. Thus, we have removed these data from the manuscript and repeated the same iMO recruitment experiments in all mouse genotypes at all time points (now shown in Figure 2 and 3). We find these data to be more consistent allowing us to strongly conclude that CCR2 and CCR7 act synergistically to control iMO recruitment during encephalitic virus infection.

5. It seems that the findings are more useful in showing the complementary roles of CCR2 and CCR7 in monocyte egress from bone marrow than the actual control of viral pathogenesis in brain. Therefore, the introduction and discussion sections should be revised to highlight this crucial aspect of their findings.

We appreciate the reviewer's comments and agree that this is a primary finding of this work. We have greatly revised the abstract, results and discussion to highlight these findings. However, with our new RNA-seq data we have elucidated multiple factors contributing to LACV iMOs inability to control viral pathogenesis in the brain. We have integrated these new findings and our interpretations into all sections of the manuscript and believe they provide substantial insight into LACV pathogenesis.

Reviewer #3 (Remarks to the Author):

The paper "CCR7 plays a complementary role to CCR2 during monocyte recruitment from the bone marrow" by Winkler et al investigate the roles of both CCR2 and CCR7 in iMO recruitment to the brain in LACV versus HSV infection and qualified the roles for both receptors. The experiments use various mouse strains and are carefully constructed. The methods used are sound and the results are novel.

We appreciate the reviewer's comments.

Negative points are: The manuscript should be edited for better understanding. Introduction and abstract could be clearer.

We have extensively edited and added to the manuscript both to incorporate new data and to clarify multiple points (please see highlighted areas).

The two ko strains used were sourced from Jackson but the original publications would be helpful to know.

We have included the original references regarding the source of the CCR2 and CCR7 knockout strains (current line 322).

Reviewers' comments:

Reviewer #1 (Remarks to the Author):

The authors present a revised manuscript entitled "CCR2 and CCR7 synergistically control inflammatory monocyte recruitment but the infecting virus dictates monocyte function in the brain". Overall, the manuscript is greatly improved, and the authors have addressed most of my questions. However, I still have one concern related with Figure 1f. In Figure 1f, few cells were analyzed and antibody control sample was not included. Furthermore, details regarding the CCR7 antibody should be included in Table 2.

Reviewer #2 (Remarks to the Author):

No comment provided. Reviewer #2 is satisfied with the revised manuscript.

Reviewers' comments:

Reviewer #1 (Remarks to the Author):

The authors present a revised manuscript entitled "CCR2 and CCR7 synergistically control inflammatory monocyte recruitment but the infecting virus dictates monocyte function in the brain". Overall, the manuscript is greatly improved, and the authors have addressed most of my questions. However, I still have one concern related with Figure 1f. In Figure 1f, few cells were analyzed and antibody control sample was not included. Furthermore, details regarding the CCR7 antibody should be included in Table 2.

We appreciate the reviewer's input and comments. We thank them for spending their valuable time on this review.

Regarding the number of iMOs analyzed from brain, for HSV between 896 and 2766 iMOs were analyzed for each sample and for LACV between 2193 and 10688 iMOs were analyzed. These numbers are in line with what we commonly observe for iMO infiltration into the brain of mice infected with either virus as infiltration can be variable. This is a sufficient number for quantification and to make relevant comparisons of CCR7 expression on recruited iMOs and to accurately reflect the degree of that expression in vivo.

Regarding the use of a control antibody in Figure 1F, we controlled for non-specific Fc receptor binding by the application of CD16/CD32 FcγIII/II blocking antibody, which is the primary purpose of an isotype control. We also used unlabeled cell, single antibody and fluorescence minus one antibody controls which allow us to confirm CCR7 antibody specificity and ensure that directly conjugated fluorochrome is not bleeding into other channels. Use of non-specific isotype controls can lead to erroneous background signal because it is impossible to know what off-target binding will occur with a given control antibody (Andersen et al¹). Lot variability, which is common for all vendors, further confounds this issue. Thus, we and others find that utilizing fluorescence minus one controls have higher accuracy and reliability in defining antibody-specific labeling than isotype controls. Additionally, the finding that iMOs from LACV infection have higher expression of CCR7 is further confirmation of the specificity of the CCR7 antibody because this finding directly correlates with transcriptomic data (Figure 1e) showing higher expression in iMOs from LACV-infected mice.

Lastly, we appreciate the reviewer catching our omission of information for the CCR7 antibody in Table 2. We also omitted information for two other antibodies, 1) and identical clone of Ly6G with a different conjugate (BV395) that allowed us to include the CCR7-BV421 in our flow antibody panel and 2) cKit which was used to identify bone marrow progenitor cells. Information on these antibodies has been added to Table 2. We have also included updated methods for the processing and labeling of bone marrow cells (lines 338-348).

Reviewer #2 (Remarks to the Author):

No comment provided. Reviewer #2 is satisfied with the revised manuscript.

We thank the reviewer and appreciate their time and input.

1. Andersen MN, Al-Karradi SN, Kragstrup TW, Hokland M. Elimination of erroneous results in flow cytometry caused by antibody binding to Fc receptors on human monocytes and macrophages. *Cytometry A*. 2016;89(11):1001-1009.